# Stationary distribution of a reaction-diffusion hepatitis B virus infection model driven by the Ornstein-Uhlenbeck process

Zhenyu Zhang[1], Guizhen Liang [2]*, Kangkang Chang[2]

1 Academy of Fine Arts, Xinxiang University, Xinxiang, P.R. China, 2 School of Mathematics and Statistics, Xinxiang University, Xinxiang, P.R. China

* lgz3361@xxu.edu.cn

**Data Availability Statement:** All relevant data are within the paper.

## Abstract

A reaction-diffusion hepatitis B virus (HBV) infection model based on the mean-reverting Ornstein-Uhlenbeck process is studied in this paper. We demonstrate the existence and uniqueness of the positive solution by constructing the Lyapunov function. The adequate conditions for the solution's stationary distribution were described. Last but not least, the numerical simulation demonstrated that reversion rates and noise intensity influenced the disease and that there was a stationary distribution. It was concluded that the solution tends more toward the stationary distribution, the greater the reversion rates and the smaller the noise.

## 1. Introduction

The hepatitis B virus is the cause of the potentially fatal liver infection known as hepatitis B. According to the World Health Organization, we knew that the first case of acute hepatitis of unknown cause was reported in the UK on 15 April 2022. Two hundred twenty-eight children in at least 20 countries had developed liver disease by 5 May [1]. They estimated that 296 million people were lived with chronic hepatitis B infection in 2019, with about 820,000 deaths [2]. Notwithstanding the accessibility of a profoundly viable immunization, around 1.5 million individuals are recently contaminated yearly [2]. Based on the above analysis, we understood that HBV still threatens human public health. Therefore, it is important to investigate the hepatitis B virus's dynamic behavior.

Mathematical models are regarded as an efficient method when it comes to comprehending how HBV is transmitted. In the meantime, much research has been done on the HBV infection model's dynamic behavior [3–12]. For example, Din and Li [6] built a stochastic HBV model with Markov switching and white noise, and verified the theorem results using Runge-Kutta method. White noise plays an important role in infection control, according to reference [8] which looked at the effect of delay on HBV recurrence and reinfection. Rihan and Alsakaj looked into how a stochastic HBV model affected the persistence of the disease and the possibility of its extinction. Ge et al. [11] solved the Foker-Planck equation. In addition, the probability density function of a stochastic HBV model close to a singular local quasi-equilibrium

**Funding:** This research was supported in part by the Startup Foundation for Doctors of Xinxiang University (No.1366020229).

**Competing interests:** The authors have declared that no competing interests exist.

was expressed specifically. The theoretical results are verified by numerical simulation. They are consistent with the HBV epidemic data in China.

We noted that the transmission of the hepatitis B virus is related to random environmental factors and the spatial location of the virus and cells [13–17]. In [13, 14], using the following model to investigate HBV's dynamics:

$$\begin{cases} \dfrac{\partial u_1}{\partial t} = \lambda(x) - a(x)u_1 - \beta(x)u_1 u_3 & x \in \Omega, t > 0 \\[2mm] \dfrac{\partial u_2}{\partial t} = \beta(x)u_1 u_3 - b(x)u_2 & x \in \Omega, t > 0 \\[2mm] \dfrac{\partial u_3}{\partial t} = d \triangle u_3 + k(x)u_2 - m(x)u_3 & x \in \Omega, t > 0, \end{cases} \tag{1}$$

where $u_1(x, t)$, $u_2(x, t)$ and $u_3(x, t)$ represent the concentration of uninfected cells, infected cells and virus, at location x and time t. $\lambda(x)$ represents the production rate of uninfected cells. $a(x)$ is the death rate of uninfected cells. Uninfected cells become infected cells at rate $\beta(x)$ $u_1 u_3$. Infected cells are produced at rate $\beta(x)u_1 u_3$. $b(x)$ is the death rate of infected cells. $k(x)$ is virus production rate. $m(x)$ is the death rate of viruses. Wu and Zou [16], in contrast to references [13, 14], focused on the diffusion of cells rather than viruses. Issa et al. [17] did not consider the spatial heterogeneity of coefficients but did consider the diffusion of viruses and cells. However, Allen [18] compared the difference between the Gaussian white noise process and the mean-reverting Ornstein-Uhlenbeck processes. The result showed that the mean-reverting Ornstein-Uhlenbeck process has better characteristics than white noise, which can describe the environmental change in biological systems well and be closer to reality theoretically and biologically. Meanwhile, the mean-reverting process is continuous, non-negative, practical and asymptotic distribution. Our simulation results also showed that as the reversion rate increases, the solution of the model is closer to the asymptotic distribution. This strategy has been generally utilized in epidemiology [19–21] and the financial economy [22, 23].

The following are the primary goals of this study: (1) By introducing cell diffusion and the mean-reverting Ornstein-Uhlenbeck process, we built the reaction-diffusion model of HBV infection. (2) The existence and uniqueness of the solution of the model and the stability of the model are proved. (3) The numerical simulation demonstrated the stationary distribution's existence and the disease's influence on reversion rates and noise intensity. It was concluded that the solution tends more toward the stationary distribution, the higher the reversion rate and the lower the noise.

The article's structure is as follows: In Section 2, the mean-reverting Ornstein-Uhlenbeck process was incorporated into the diffusion HBV infection model. In Section 3, we proved the existence and uniqueness of the solution. Then, sufficient conditions are given for the diffusion HBV infection model. Numerical simulation is provided in Section 4 to demonstrate the theoretical findings. The conclusion is made in Section 5.

## 2. Model

We consider the following model:

$$\begin{cases} \dfrac{\partial u_1}{\partial t} = d_1 \triangle u_1 + \lambda(x) - a(x)u_1 - \beta(x)u_1 u_3, & x \in \Omega, t > 0, \\[2mm] \dfrac{\partial u_2}{\partial t} = d_2 \triangle u_2 + \beta(x)u_1 u_3 - b(x)u_2, & x \in \Omega, t > 0, \\[2mm] \dfrac{\partial u_3}{\partial t} = d_3 \triangle u_3 + k(x)u_2 - m(x)u_3, & x \in \Omega, t > 0, \end{cases} \tag{2}$$

with boundary condition

$$\frac{\partial u_1}{\partial n} = \frac{\partial u_2}{\partial n} = \frac{\partial u_3}{\partial n} = 0, \ x \in \partial\Omega, \ t > 0, \tag{3}$$

and initial condition

$$u_1(x, 0) = u_{10}(x), u_2(x, 0) = u_{20}(x), u_3(x, 0) = u_{30}(x), x \in \Omega. \tag{4}$$

The effects of a random environment are not considered in the above model. Furthermore, we introduce the mean-reverting Ornstein-Uhlenbeck process, which has the following form:

$$\begin{cases} da(x, t) = \vartheta_1(a_e - a(t))dt + \varepsilon_1 dB_1(t), \\ db(x, t) = \vartheta_2(b_e - b(t))dt + \varepsilon_2 dB_2(t), \\ dm(x, t) = \vartheta_3(m_e - m(t))dt + \varepsilon_3 dB_3(t), \end{cases} \tag{5}$$

where $\vartheta_i, \varepsilon_i$ and $B_i(t), (i = 1, 2, 3)$ represent the reversion rates, noise intensity, are Brownian motion, respectively.

The stochastic integral format for the arithmetic Ornstein-Uhlenbeck process (5) enables us to obtain the following explicit form solution:

$$\begin{cases} a(t) = a_e + (a_0 - a_e)e^{-\vartheta_1 t} + \varepsilon_1 \int_0^t e^{-\vartheta_1(t-s)}dB_1(s), \\ b(t) = b_e + (b_0 - b_e)e^{-\vartheta_2 t} + \varepsilon_2 \int_0^t e^{-\vartheta_2(t-s)}dB_2(s), \\ m(t) = m_e + (m_0 - m_e)e^{-\vartheta_3 t} + \varepsilon_3 \int_0^t e^{-\vartheta_3(t-s)}dB_3(s). \end{cases} \tag{6}$$

By [20], Eq (6) can be almost surely (a.s.) rewritten as:

$$\begin{cases} a(t) = a_e + (a_0 - a_e)e^{-\vartheta_1 t} + \xi_1(t)\frac{dB_1(t)}{dt}, \\ b(t) = b_e + (b_0 - b_e)e^{-\vartheta_2 t} + \xi_2(t)\frac{dB_2(t)}{dt}, \\ m(t) = m_e + (m_0 - m_e)e^{-\vartheta_3 t} + \xi_3(t)\frac{dB_3(t)}{dt}, \end{cases} \tag{7}$$

where $a_0 := a(0) > 0, b_0 := b(0) > 0, m_0 := m(0) > 0, \xi_i = \frac{\varepsilon_i}{2\vartheta_i}\sqrt{1 - e^{-2\vartheta_i t}}, (i = 1, 2, 3)$. Substituting (7) into system (2) implies the following stochastic system

$$\begin{cases} du_1 = [d_1 \triangle u_1 + \lambda(x) - a_e u_1 - (a_0 - a_e)e^{-\vartheta_1(t)}u_1 - \beta(x)u_1 u_3]dt - \xi_1(t)u_1 dB_1(t), \\ du_2 = [d_2 \triangle u_2 + \beta(x)u_1 u_3 - b_e u_2 - (b_0 - b_e)e^{-\vartheta_2 t}u_2]dt - \xi_2(t)u_2 dB_2(t), \\ du_3 = [d_3 \triangle u_3 + k(x)u_2 - m_e u_3 - (m_0 - m_e)e^{-\vartheta_3 t}u_3]dt - \xi_3(t)u_3 dB_3(t), \end{cases} \tag{8}$$

with boundary condition

$$\frac{\partial u_1}{\partial n} = \frac{\partial u_2}{\partial n} = \frac{\partial u_3}{\partial n} = 0, x \in \partial\Omega, t > 0,$$

and initial condition

$$u_1(x, 0) = u_{10}(x), u_2(x, 0) = u_{20}(x), u_3(x, 0) = u_{30}(x), x \in \Omega.$$

Let $B$ be a linear operator defined by

$$\mathbb{B}\begin{pmatrix} u_1 \\ u_2 \\ u_3 \end{pmatrix} = \begin{pmatrix} d_1 \triangle u_1 \\ d_2 \triangle u_2 \\ d_3 \triangle u_3 \end{pmatrix}. \tag{9}$$

Then, we define a nonlinear operator $C$ by

$$\mathbb{C}\begin{pmatrix} u_1 \\ u_2 \\ u_3 \end{pmatrix} = \begin{pmatrix} \lambda(x) - a_e u_1 - (a_0 - a_e)e^{-\vartheta_1(t)}u_1 - \beta(x)u_1 u_3 - \xi_1(t)u_1\dot{B}_1(t) \\ \beta(x)u_1 u_3 - b_e u_2 - (b_0 - b_e)e^{-\vartheta_2 t}u_2 - \xi_2(t)u_2\dot{B}_2(t) \\ k(x)u_2 - m_e u_3 - (m_0 - m_e)e^{-\vartheta_3 t}u_3 - \xi_3(t)u_3\dot{B}_3(t) \end{pmatrix}. \tag{10}$$

Let $\mathcal{W}(t) = (u_1(x,t), u_2(x,t), u_3(x,t))^T$, together with Eqs (9) and (10), system (8) has been rewritten as the following abstract Cauchy problem

$$\frac{d}{dt}\mathcal{W}(t) = \mathbb{B}\mathcal{W}(t) + \mathbb{C}\mathcal{W}(t). \tag{11}$$

## 3. Main result

### 3.1. Existence and unique of solution

Let $(\Omega, \mathcal{F}, \{\mathcal{F}_t\}_{t\geq 0}, P)$ be a complete probability space with a filtration $\{\mathcal{F}_t\}_{t\geq 0}$, and $B_i(t)$, $(i = 1, 2, 3)$ defined on $(\Omega, \mathcal{F}, \{\mathcal{F}_t\}_{t\geq 0}, P)$, $R_+^3 = (x_1, x_2, x_3) \in R^3, x_i > 0, (i = 1, 2, 3)$. Next, we introduce a lemma that gives a criterion for the existence of an ergodic stationary distribution to system (8).

Notation

$$\bar{g} = \sup_{t\to\infty} g(t), \underline{g} = \inf_{t\to\infty} g(t), \tag{12}$$

here, $g(t)$is a continuous bounded function.

**Lemma 3.1**. *For any initial data* $(u_{10}, u_{20}, u_{30})$, *the solution* $u(x, t) = (u_1(x, t), u_2(x, t), u_3(x, t))$ *of system* (8), *satisfies that*

$$\limsup_{t\to\infty}(E\|u_1(x,t)\| + E\|u_2(x,t)\| + E\|u_3(x,t)\|) < M_1,$$

*where* $M_1$ *is a positive constant.*

*Proof.* Let

$$\mathbb{N}(t) = \int_\Omega E[ku_1(x,t) + ku_2(x,t) + b_e u_3(x,t)]dx,$$

by (8), we have

$$
\begin{aligned}
\frac{dN(t)}{dt} \quad &= \int_\Omega E\left[k\frac{\partial}{\partial t}u_1(x,t) + k\frac{\partial}{\partial t}u_2(x,t) + b_e\frac{\partial}{\partial t}u_3(x,t)\right]dx \\
&= \int_\Omega E[kd_1 \triangle u_1(x,t) + k\lambda - ka_e u_1(x,t) - k(a_0 - a_e)e^{-\vartheta_1(t)}u_1(x,t) - k\beta(x)u_1(x,t)u_3(x,t) \\
&\quad - k\xi_1(t)u_1(x,t)\dot{B}_1(t) + kd_2 \triangle u_2(x,t) + k\beta(x)u_1(x,t)u_3(x,t) - kb_e u_2(x,t) - k(b_0 - b_e)e^{-\vartheta_2 t}u_2(x,t) \\
&\quad - k\xi_2(t)u_2(x,t)\dot{B}_2(t) + b_e d_3 \triangle u_3(x,t) + b_e k(x)u_2(x,t) - b_e m_e u_3(x,t) - b_e(m_0 - m_e)e^{-\vartheta_3 t}u_3(x,t) \\
&\quad - b_e\xi_3(t)u_3(x,t)\dot{B}_3(t)] \\
&\le kd_1 \int_\Omega E(\triangle u_1(x,t))dx + kd_2 \int_\Omega E(\triangle u_2(x,t))dx + b_e d_3 \int_\Omega E(\triangle u_3(x,t)) \\
&\quad + \int_\Omega E(k\lambda - ka_e(1 - e^{-\vartheta_1 t})u_1(x,t) - kb_e(1 - e^{-\vartheta_2 t})u_2(x,t) - b_e m_e(1 - e^{-\vartheta_3 t})u_3(x,t))dx \\
&\le kd_1 \int_{\partial\Omega} E(\frac{\partial}{\partial n}u_1(x,t))dx + kd_2 \int_{\partial\Omega} E(\frac{\partial}{\partial n}u_2(x,t))dx + b_e d_3 \int_{\partial\Omega} E(\frac{\partial}{\partial n}u_3(x,t))dx \\
&\quad + \int_\Omega E(k\lambda - A(ku_1(x,t) + ku_2(x,t) + b_e m_e u_3(x,t)))dx \\
&= k\lambda|\Omega| - A\mathbb{N}(t),
\end{aligned}
$$

where $|\Omega|$ denotes the volume of $\Omega$, $A = min\{a_e(1 - e^{-\vartheta_1 t}), b_e(1 - e^{-\vartheta_2 t}), m_e(1 - e^{-\vartheta_3 t})\}$. This implies that

$$
\lim_{t\to+\infty} \mathbb{N}(t) \le \frac{k\lambda|\Omega|}{A} := M_1.
$$

**Remark 1** Lemma 3.1 means that the solution is boundness for system (8).
Furthermore, we prove the existence and unique of solution.

**Theorem 3.2** *For any initial data $(u_{10}, u_{20}, u_{30}) > 0$, there exists a unique solution $(u_1(x,t), u_2(x,t), u_3(x,t)) > 0$ of system (8) for $t > 0$ on $\Omega$.*

*Proof.* Since the coefficients of system (8) satisfy the local Lipschitz condition, there is a unique local solution on $t \in [0, \tau_e)$, where $\tau_e$ is the explosion time Let $l_0 > 0$ be sufficiently large for

$$
\frac{1}{l_0} \le \min_{0 < t < \tau_e} |\mathcal{W}(t)| \le \max_{0 < t < \tau_e} |\mathcal{W}(t)| \le l_0.
$$

For each integer $l > l_0$, define the stopping time

$$
\tau_l = inf\{t \in [0, \tau_e] : \min(u_1, u_2, u_3) \le \frac{1}{l} \text{ or } \max(u_1, u_2, u_3) \ge l\}.
$$

Let $inf\,\emptyset = \infty$ ($\emptyset$ represents the empty set). $\tau_l$ is increasing as $l \to \infty$. Let $\tau_\infty = \lim_{l\to\infty} \tau_l$, then $\tau_\infty < \tau_e$ a.s. In the following, we need to show $\tau_\infty = \infty$ a.s. Therefore, according to Itô's

formula, we have

$$
\begin{aligned}
d(\|u_1(x,t)\|^2 &+ \|u_2(x,t)\|^2 + \|u_3(x,t)\|^2) \\
&= \{2\langle u_1(x,t), d_1 \triangle u_1 + \lambda(x) - a_e u_1 - (a_0 - a_e)e^{-\vartheta_1(t)}u_1 - \beta(x)u_1 u_3\rangle \\
&\quad + 2\langle u_2(x,t), d_2 \triangle u_2 + \beta(x)u_1 u_3 - b_e u_2 - (b_0 - b_e)e^{-\vartheta_2 t}u_2\rangle \\
&\quad + 2\langle u_3(x,t), d_3 \triangle u_3 + k(x)u_2 - m_e u_3 - (m_0 - m_e)e^{-\vartheta_3 t}u_3\rangle + \xi_1^2(t)\|u_1(x,t)\|^2 \\
&\quad + \xi_2^2(t)\|u_2(x,t)\|^2 + \varepsilon_3^2(t)\|u_3(x,t)\|^2\}dt + 2\langle u_1(x,t), -\xi_1 u_1(x,t)dB_1(t)\rangle \\
&\quad + 2\langle u_2(x,t), -\xi_2 u_2(x,t)dB_2(t)\rangle + 2\langle u_3(x,t), -\xi_3 u_3(x,t)dB_3(t)\rangle.
\end{aligned}
\tag{13}
$$

Now, let $l > l_0$ and $T > 0$, we can integrate both sides of (13) from 0 to $\tau_l \wedge T$ and then take the expectations to get

$$
E[\|u_1(x, \tau_l \wedge T)\|^2 + \|u_2(x, \tau_l \wedge T)\|^2 + \|u_3(x, \tau_l \wedge T)\|^2] - (\|u_{10}\|^2 + \|u_{20}\|^2 + \|u_{30}\|^2)
$$

$$
= E \int_0^{\tau_l \wedge T} \{-2d_1\|\nabla u_1(x,s)\|^2 + 2\langle u_1(x,s), \lambda\rangle - 2a_e\|u_1(x,s)\|^2 - 2(a_0 - a_e)e^{-\vartheta_1 t}\|u_1(x,s)\|^2
$$

$$
-2\langle u_1(x,s), \beta u_1(x,s)u_3(x,s)\rangle - 2d_2\|\nabla u_2(x,s)\|^2 + 2\langle u_2(x,s), \beta u_1(x,s)u_3(x,s)\rangle - 2b_e\|u_2(x,s)\|^2
$$

$$
-2(b_0 - b_e)e^{-\vartheta_2 t}\|u_2(x,s)\|^2 - 2d_3\|\nabla u_3(x,s)\|^2 + 2\langle u_3(x,s), ku_2(x,s)\rangle - 2m_e\|u_3(x,s)\|^2 -
$$

$$
2(m_0 - m_e)e^{-\vartheta_3 t}\|u_3(x,s)\|^2 + \xi_1(s)^2\|u_1(x,s)\|^2 + \xi_2(s)^2 u_2(x,s) + \xi_3(s)^2 u_3(x,s)\}ds
$$

$$
\leq E \int_0^{\tau_l \wedge T} \{2\langle u_1(x,s), \lambda\rangle + 2\langle u_2(x,s), \beta u_1(x,s)u_3(x,s)\rangle + 2\langle u_3(x,s), ku_2(x,s)\rangle + \xi_1^2(s)\|u_1(x,s)\|^2 +
$$

$$
\xi_2^2(s)u_2(x,s) + \xi_3^2(s)u_3(x,s)\}ds.
$$

Then according to Lemma 3.1 and fundamental inequality, we have

$$
\begin{aligned}
E \quad & [\|u_1(x, \tau_l \wedge T)\|^2 + \|u_2(x, \tau_l \wedge T)\|^2 + \|u_3(x, \tau_l \wedge T)\|^2] \\
&\leq (\|u_{10}\|^2 + \|u_{20}\|^2 + \|u_{30}\|^2) + E \int_0^{\tau_l \wedge T} \{\|u_1(x,s)\|^2 + \bar{\lambda}^2 + \|u_2(x,s)\|^2 + \beta^2 M_1^2\|u_3(x,s)\|^2 \\
&\quad + \|u_3(x,s)\|^2 + k^2\|u_2(x,s)\|^2 + \xi_1^2(s)\|u_1(x,s)\|^2 + \xi_2^2(s)u_2(x,s) + \xi_3^2(s)u_3(x,s)\}ds \\
&\leq M_2 + M_3 E \int_0^{\tau_l \wedge T} \{\|u_1(x,s)\|^2 + \|u_2(x,s)\|^2 + \|u_3(x,s)\|^2\}ds,
\end{aligned}
$$

where

$$
M_2 = \|u_{10}\|^2 + \|u_{20}\|^2 + \|u_{30}\|^2 + \bar{\lambda}^2 \tau_l,
$$

$$
M_3 = max\{(1 + \xi_1^2(s)), (1 + k^2 + \xi_2^2(s)), (1 + \beta^2 M_1^2 + \xi_3^2(s))\}.
$$

By the Gronwall inequality, we have

$$
E[\|u_1(x, \tau_l \wedge T)\|^2 + \|u_2(x, \tau_l \wedge T)\|^2 + \|u_3(x, \tau_l \wedge T)\|^2] \leq M_2 e^{M_3 T}.
\tag{14}
$$

Define

$$\lambda_l = \inf_{\|\mathcal{W}(t)\|>l, 0<t<\infty} (\|u_1(x,t)\|^2 + \|u_2(x,t)\|^2 + \|u_3(x,t)\|^2), \textit{for any } l > l_0. \tag{15}$$

Combine (14) and (15) to get

$$\lambda_l P(\tau_l \leq T) \leq M_2 e^{M_3 T},$$

since $\lim_{l\to\infty} \lambda_l = \infty$, in the above inequality, let $l \to \infty$, we can get $P(\tau_\infty \leq T) = 0$, namely,

$$P(\tau_l \geq T) = 1.$$

By (14), $l \to \infty$ means that

$$E[\|u_1(x,T)\|^2 + \|u_2(x,T)\|^2 + \|u_3(x,T)\|^2] \leq M_2 e^{M_3 T}.$$

This proof is complete. The above theorem represents the system (8) exists a unique global solution.

**Remark 2** Theorem 3.2 represents the system (8) exists a unique global solution.

**Theorem 3.3** *With respect to the function $V = \|u_1(x,t)\|^2 + \|u_2(x,t)\|^2 + \|u_3(x,t)\|^2$, we have*

$$\limsup_{t\to\infty} \frac{1}{t} ln(E(\|u_1(x,t)\|^2 + \|u_2(x,t)\|^2 + \|u_3(x,t)\|^2)) \leq M_3.$$

*Proof.* By virtue of Eq (13), we have $V \leq -\frac{\bar{\lambda}^2}{M_3} + ce^{M_3 t}$ (where $c > 0$ is a constant). Moreover, we will prove the bounded of $LV$, according to the Eq (13), we can obtain

$$
\begin{aligned}
LV = \ & 2\langle u_1(x,t), d_1 \triangle u_1 + \lambda(x) - a_e u_1 - (a_0 - a_e)e^{-\vartheta_1(t)}u_1 - \beta(x)u_1 u_3\rangle \\
& + 2\langle u_2(x,t), d_2 \triangle u_2 + \beta(x)u_1 u_3 - b_e u_2 - (b_0 - b_e)e^{-\vartheta_2 t}u_2\rangle \\
& + 2\langle u_3(x,t), d_3 \triangle u_3 + k(x)u_2 - m_e u_3 - (m_0 - m_e)e^{-\vartheta_3 t}u_3\rangle \\
& + \xi_1^2(t)\|u_1(x,t)\|^2 + \xi_2^2(t)\|u_2(x,t)\|^2 + \varepsilon_3^2(t)\|u_3(x,t)\|^2 \\
= \ & -2d_1\|\nabla u_1(x,s)\|^2 + 2\langle u_1(x,s), \lambda\rangle - 2a_e\|u_1(x,s)\|^2 - 2(a_0 - a_e)e^{-\vartheta_1 t}\|u_1(x,s)\|^2 \\
& -2\langle u_1(x,s), \beta u_1(x,s)u_3(x,s)\rangle - 2d_2\|\nabla u_2(x,s)\|^2 + 2\langle u_2(x,s), \beta u_1(x,s)u_3(x,s)\rangle - 2b_e\|u_2(x,s)\|^2 \\
& -2(b_0 - b_e)e^{-\vartheta_2 t}\|u_2(x,s)\|^2 - 2d_3\|\nabla u_3(x,s)\|^2 + 2\langle u_3(x,s), ku_2(x,s)\rangle - 2m_e\|u_3(x,s)\|^2 - \\
& 2(m_0 - m_e)e^{-\vartheta_3 t}\|u_3(x,s)\|^2 + \xi_1(s)^2\|u_1(x,s)\|^2 + \xi_2(s)^2 u_2(x,s) + \xi_3(s)^2 u_3(x,s)\}ds \\
\leq \ & \bar{\lambda}^2 + M_3(\|u_1(x,t)\|^2 + \|u_2(x,s)\|^2 + \|u_3(x,s)\|^2).
\end{aligned}
$$

For $V$, using Itô's formula:

$$
\begin{aligned}
EV \ &= (\|u_{10}\|^2 + \|u_{20}\|^2 + \|u_{30}\|^2) + E\int_0^t LVds \\
&\leq (\|u_{10}\|^2 + \|u_{20}\|^2 + \|u_{30}\|^2) + \bar{\lambda}^2 t + M_3 E\int_0^t ((\|u_1(x,t)\|^2 + \|u_2(x,s)\|^2 + \|u_3(x,s)\|^2))ds \\
&\leq (\|u_{10}\|^2 + \|u_{20}\|^2 + \|u_{30}\|^2) + \bar{\lambda}^2 t + M_3 E\int_0^t (-\frac{\bar{\lambda}^2}{M_3} + ce^{M_3 t})ds.
\end{aligned}
$$

According to the arbitrariness of $c$, we have

$$EV \leq \frac{ce^{M_3 t}}{M_3}.$$

The result of the theorem can be obtained.

**Remark 3** Theorem 3.3 denotes the square exponent stability of the Lyapunov function.

**Theorem 3.4**. *If $E(\|u_{10}\|^2 + \|u_{20}\|^2 + \|u_{30}\|^2) \leq Z_1$, we have*

$$E(\|u_1(x, t)\|^2 + \|u_2(x, t)\|^2 + \|u_3(x, t)\|^2) \leq Z_2, \quad t \in [0, T],$$

*where $Z_1$, $Z_2$, $T$ are positive real numbers. Then system* (8) *is finite-time stable.*

*Proof.* According to Theorem 3.2, we can obtain the proof of the theorem.

**Remark 4** Theorem 3.4 denotes the model is finite-time stable.

Next, we prove the stationary distribution of the solution for system (8).

## 3.2. Stationary distribution of solution

First, we introduce the follow theorem.

**Theorem 3.5** *For any $\kappa > 0$, we have*

$$E(\sup_{0 \leq t \leq T} \|u_1(x, t)\|^k + \sup_{0 \leq t \leq T} \|u_1(x, t)\|^k + \sup_{0 \leq t \leq T} \|u_1(x, t)\|^k) \leq M_\kappa,$$

*where $M_\kappa$ is a constant that depends only on $\kappa$.*

*Proof.* First, we consider $\kappa > 1$, By applying the Itô's formula, we have

$$\|u_1(x, s)\|^\kappa + \|u_2(x, s)\|^\kappa + \|u_3(x, s)\|^\kappa - \|u_{10}\|^\kappa - \|u_{20}\|^\kappa - \|u_{30}\|^\kappa$$

$$= \int_0^t \{\kappa \|u_1(x, s)\|^{\kappa-2} \langle u_1(x, s), d_1 \triangle u_1 + \lambda(x) - a_e u_1 - (a_0 - a_e)e^{-\vartheta_1(t)} u_1 - \beta(x) u_1 u_3 \rangle$$

$$+ \kappa \|u_2(x, s)\|^{\kappa-2} \langle u_2(x, s), d_2 \triangle u_2 + \beta(x) u_1 u_3 - b_e u_2 - (b_0 - b_e)e^{-\vartheta_2 t} u_2 \rangle + \kappa \|u_3(x, s)\|^{\kappa-2}$$

$$\langle u_3(x, s), d_3 \triangle u_3 + k(x) u_2 - m_e u_3 - (m_0 - m_e)e^{-\vartheta_3 t} u_3 \rangle + \frac{1}{2}\kappa(\kappa-1)\xi_1^2(s)\|u_1(x, s)\|^2$$

$$+ \frac{1}{2}\kappa(\kappa-1)\xi_2^2(s)\|u_2(x, s)^2\| + \frac{1}{2}\kappa(\kappa-1)\xi_3^2(s)\|u_3(x, s)^2\|\}ds - \int_0^t \kappa \xi_1(s)\|u_1(x, s)\|^\kappa dB_1(s)$$

$$- \int_0^t \kappa \xi_2(s)\|u_2(x, s)\|^\kappa dB_2(s) - \int_0^t \kappa \xi_3(s)\|u_3(x, s)\|^\kappa dB_1(s).$$

Next, we take the $sup(\cdot)$ and expectation of the above equation

$$E \sup_{0 \le t \le T} \{\|u_1(x,s)\|^\kappa + \|u_2(x,s)\|^\kappa + \|u_3(x,s)\|^\kappa\}$$

$$\le E \sup_{0 \le t \le T} (\|u_{10}\|^\kappa + \|u_{20}\|^\kappa + \|u_{30}\|^\kappa) + E \sup_{0 \le t \le T} \int_0^t \{-d_1\kappa\|u_1(x,s)\|^{\kappa-2}\|\nabla u_1(x,s)\|^2$$

$$+\kappa\lambda\|u_1(x,s)\|^{\kappa-1} - \kappa a_e\|u_1(x,s)\|^\kappa - \kappa(a_0 - a_e)e^{-\vartheta_1 t}\|u_1(x,s)\|^\kappa - \kappa\beta\|u_1(x,s)\|^2$$

$$\langle u_1(x,s), u_1(x,s)u_3(x,s)\rangle - d_2\kappa\|u_2(x,s)\|^{\kappa-2}\|\nabla u_2(x,s)\|^2 - \kappa b_e\|u_2(x,s)\|^2$$

$$+\kappa\beta\|u_2(x,s)\|^{\kappa-2}\langle u_2(x,s), u_1(x,s)u_3(x,s)\rangle - \kappa(b_0 - b_e)e^{-\vartheta_3 t}\|u_2(x,s)\|^2 - d_3\kappa\|u_3(x,s)\|^{\kappa-2}$$

$$\|\nabla u_1(x,s)\|^2 + \kappa k\|u_3(x,s)\|^{\kappa-2}\langle u_3(x,s), u_2(x,s)\rangle - \kappa m_e\|u_3(x,s)\|^\kappa - \kappa(m_0 - m_e)e^{-\vartheta_3 t}$$

$$\|u_3(x,s)\|^\kappa + \frac{1}{2}\kappa(\kappa-1)\xi_1^2(s)\|u_1(x,s)\|^2 + \frac{1}{2}\kappa(\kappa-1)\xi_2^2(s)\|u_2(x,s)\|^2 + \frac{1}{2}\kappa(\kappa-1)\xi_3^2(s)$$

$$\|u_3(x,s)\|^2\}ds - E\sup_{0 \le t \le T}\int_0^t \kappa\xi_1(s)\|u_1(x,s)\|^\kappa dB_1(s) - E\sup_{0 \le t \le T}\int_0^t \kappa\xi_2(s)\|u_2(x,s)\|^\kappa dB_2(s)$$

$$-E\sup_{0 \le t \le T}\int_0^t \kappa\xi_3(s)\|u_3(x,s)\|^\kappa dB_3(s)$$

$$\le E\sup_{0 \le t \le T}(\|u_{10}\|^\kappa + \|u_{20}\|^\kappa + \|u_{30}\|^\kappa) + E\sup_{0 \le t \le T}\int_0^t\{\kappa\lambda\|u_1(x,s)\|^{\kappa-1} + \kappa\beta\|u_2(x,s)\|^{\kappa-2}$$

$$\langle u_2(x,s), u_1(x,s)u_3(x,s)\rangle + \kappa k\|u_3(x,s)\|^{\kappa-2}\langle u_3(x,s), u_2(x,s)\rangle + \frac{1}{2}\kappa(\kappa-1)$$

$$\xi_1^2(s)\|u_1(x,s)\|^2 + \frac{1}{2}\kappa(\kappa-1)\xi_2^2(s)\|u_2(x,s)^2\| + \frac{1}{2}\kappa(\kappa-1)\xi_3^2(s)\|u_3(x,s)^2\|\}ds$$

$$-E\sup_{0 \le t \le T}\int_0^t \kappa\xi_1(s)\|u_1(x,s)\|^\kappa dB_1(s) - E\sup_{0 \le t \le T}\int_0^t \kappa\xi_2(s)\|u_2(x,s)\|^\kappa dB_2(s)$$

$$-E\sup_{0 \le t \le T}\int_0^t \kappa\xi_3(s)\|u_3(x,s)\|^\kappa dB_3(s).$$

Using the Young inequality and Burkholder-Davis-Gundy inequality, we have

$$E\sup_{0 \le t \le T}\{\|u_1(x,s)\|^\kappa + \|u_2(x,s)\|^\kappa + \|u_3(x,s)\|^\kappa\}$$

$$\le E(\|u_{10}\|^\kappa + \|u_{20}\|^\kappa + \|u_{30}\|^\kappa) + E\sup_{0 \le t \le T}\int_0^t\{\lambda^\kappa + (\kappa - 1 + \frac{1}{2}\kappa(\kappa-1)\xi_1^2)\|u_1(x,s)\|^2$$

$$+(\kappa - 1 + k^\kappa + \frac{1}{2}\kappa(\kappa-1)\xi_2^2)\|u_2(x,s)\|^2 + (\kappa - 1 + \beta^\kappa M_1^\kappa + \frac{1}{2}\kappa(\kappa-1)\xi_3^2)\|u_3(x,s)\|^2\}$$

$$+E\sup_{0 \le t \le T}|\int_0^t \kappa\xi_1(s)\|u_1(x,s)\|^\kappa dB_1(s)|| + E\sup_{0 \le t \le T}|\int_0^t \kappa\xi_2(s)\|u_2(x,s)\|^\kappa dB_2(s)|$$

$$+E\sup_{0 \le t \le T}|\int_0^t \kappa\xi_3(s)\|u_3(x,s)\|^\kappa dB_3(s)|$$

$$\leq E(\|u_{10}\|^{\kappa} + \|u_{20}\|^{\kappa} + \|u_{30}\|^{\kappa}) + \lambda^{\kappa}T + M_4 E \sup_{0 \leq t \leq T} \int_0^t \{\|u_1(x,s)\|^{\kappa} + \|u_2(x,s)\|^{\kappa} + \|u_3(x,s)\|^{\kappa}\}ds$$

$$+E \sup_{0 \leq t \leq T}\|u_1(x,s)\|^{\kappa/2}(\int_0^t \kappa^2 \xi_1^2(s)\|u_1(x,s)\|^2)^{1/2} + E \sup_{0 \leq t \leq T}\|u_2(x,s)\|^{\kappa/2}(\int_0^t \kappa^2 \xi_1^2(s)\|u_2(x,s)\|^2)^{1/2}$$

$$+E \sup_{0 \leq t \leq T}\|u_3(x,s)\|^{\kappa/2}(\int_0^t \kappa^2 \xi_1^2(s)\|u_3(x,s)\|^2)^{1/2}$$

$$\leq E(\|u_{10}\|^{\kappa} + \|u_{20}\|^{\kappa} + \|u_{30}\|^{\kappa}) + \lambda^{\kappa}T + M_4 E \sup_{0 \leq t \leq T} \int_0^t \{\|u_1(x,s)\|^{\kappa} + \|u_2(x,s)\|^{\kappa} + \|u_3(x,s)\|^{\kappa}\}ds$$

$$+\frac{1}{2}E \sup_{0 \leq t \leq T}(\|u_1(x,s)\|^{\kappa} + \|u_2(x,s)\|^{\kappa} + \|u_3(x,s)\|^{\kappa})$$

$$\leq 2E(\|u_{10}\|^{\kappa} + \|u_{20}\|^{\kappa} + \|u_{30}\|^{\kappa}) + 2\lambda^{\kappa}T + M_4 E \sup_{0 \leq t \leq T} \int_0^t \{\|u_1(x,s)\|^{\kappa} + \|u_2(x,s)\|^{\kappa} + \|u_3(x,s)\|^{\kappa}\}ds,$$

where
$$M_4 = max\{\kappa - 1 + \frac{1}{2}\kappa(\kappa-1)\xi_1^2, \kappa - 1 + k^{\kappa} + \frac{1}{2}\kappa(\kappa-1)\xi_2^2, \kappa - 1 + \beta^{\kappa}M_1^{\kappa} + \frac{1}{2}\kappa(\kappa-1)\xi_3^2\}.$$

According to the Gronwall inequality, we obtained

$$E \sup_{0 \leq t \leq T}\{\|u_1(x,s)\|^{\kappa} + \|u_2(x,s)\|^{\kappa} + \|u_3(x,s)\|^{\kappa}\}$$

$$\leq (2E(\|u_{10}\|^{\kappa} + \|u_{20}\|^{\kappa} + \|u_{30}\|^{\kappa}) + 2\lambda^{\kappa}T)e^{2M_4 T}$$

$$:= M_{\kappa}.$$

For $0 < \kappa < 1$, based on the Cauchy-Schwartz inequality, we obtain

$$E \sup_{0 \leq t \leq T}\{\|u_1(x,s)\|^{\kappa} + \|u_2(x,s)\|^{\kappa} + \|u_3(x,s)\|^{\kappa}\}$$

$$\leq (E1^{\frac{2}{2-\kappa}})^{1-\kappa/2}\{E(\sup_{0 \leq s \leq T}\{\|u_1(x,s)\|^{\kappa} + \|u_2(x,s)\|^{\kappa} + \|u_3(x,s)\|^{\kappa})^{\frac{2}{\kappa}}\}^{\frac{\kappa}{2}}$$

$$:= M_{\kappa}.$$

This proof is completed.

**Remark 5** Theorem 3.5 indicates that the solution of the model is $k$−moment bounded.

Next, we will give sufficient conditions for the existence and uniqueness of stationary distribution of the solution to the diffusion HBV infection model.

**Definition 3.1** [24] A stationary distribution for
$\mathcal{W}(x,t) = (u_1(x,t), u_2(x,t), u_3(x,t)), t \geq 0$, of system (8) is defined as a probability measure $\lambda \in P(\Omega)$ satisfying

$$\lambda(f) = \lambda(P_t f), \ t > 0,$$

here

$$\lambda(f) := \int_{\Omega} f(\psi)\lambda(d\psi), P_t f(\psi) := Ef(\mathcal{W}(x,t,\psi)), f \in C_b(\Omega).$$

For $\lambda_1, \lambda_2 \in P(\Omega)$, define a metric on $P(\Omega)$ by

$$d(\lambda_1, \lambda_2) = \sup_{f \in \mathcal{A}}|\int_{\Omega} f(\psi)\lambda_1(d\psi) - f(\varphi)\lambda_2(d\varphi)|$$

where

$$\mathcal{A} := \{f : \Omega \to R, |f(\psi) - f(\varphi)| \leq |\psi - \varphi|_\Omega, \psi, \varphi \in \Omega \ and \ |f(\cdot)| \leq 1\},$$

$P(\Omega)$ is complete under the metric $d(\cdot, \cdot)$. So, we have the following lemma

**Lemma 3.6** *For any bounded subset $B$ of $\Omega$, $m \geq 1$, we have*

(1) $\lim\limits_{t \to \infty} \sup\limits_{\psi, \varphi \in B} E\|\mathcal{W}(x, t, \psi) - \mathcal{W}(x, t, \varphi)\|_\Omega^m = 0$;

(2) $\lim\limits_{t \to \infty} \sup\limits_{\psi \in B} E\|\mathcal{W}(x, t, \psi)\|_\Omega^m < \infty$.

**Theorem 3.7** *For system* (8), *there exists a unique stationary distribution $\lambda \in P(\Omega)$ for* $\mathcal{W}(x, t) = (u_1(x, t), u_2(x, t), u_3(x, t)), t \geq 0$.

*Proof.* The Theorem 3.5 is equal to condition (2) in Lemma 3.6. In order to complete proof, we only need to verify that condition (1) is valid. Next, we consider the difference of two mild solutions of system (8) with distinct initial data $\psi, \varphi \in \Omega$

$$e(x, t) = \begin{pmatrix} e_1(x, t, \psi, \varphi) \\ e_2(x, t, \psi, \varphi) \\ e3(x, t, \psi, \varphi) \end{pmatrix} = \begin{pmatrix} u_1(x, t, \psi) - u_1(x, t, \varphi) \\ u_2(x, t, \psi) - u_2(x, t, \varphi) \\ u_3(x, t, \psi) - u_3(x, t, \varphi) \end{pmatrix}, \tag{16}$$

with $\|e(x, t, \psi, \varphi)\|^\kappa = \|e_1(x, t, \psi, \varphi)\|^\kappa + \|e_2(x, t, \psi, \varphi)\|^\kappa + \|e_3(x, t, \psi, \varphi)\|^\kappa$, by lemma 3.1 and Itô's formula, we have

$$d \quad (e^{\eta t}\|e(x, t, \psi, \varphi)\|^\kappa)$$

$= \eta e^{\eta t}\|e(x, t, \psi, \varphi)\|^\kappa dt + e^{\eta t}\{\kappa\|e_1(x, t, \psi, \varphi)\|^{\kappa-2}\langle e_1(x, t, \psi, \varphi), d_1 \triangle e_1(x, t, \psi, \varphi) -$

$a_e e_1(x, t, \psi, \varphi) - (a_0 - a_e)e^{-\vartheta_1 t}e_1(x, t, \psi, \varphi) - \beta(u_1(x, t, \psi)u_3(x, t, \psi) - u_1(x, t, \varphi)u_3(x, t, \varphi))\rangle dt +$

$\kappa\|e_2(x, t, \psi, \varphi)\|^{\kappa-2}\langle e_2(x, t, \psi, \varphi), d_2 \triangle e_2(x, t, \psi, \varphi) + \beta(u_1(x, t, \psi)u_3(x, t, \psi) - u_1(x, t, \varphi)u_3(x, t, \varphi))$

$-b_e e_2(x.t, \psi, \varphi) - (b_0 - b_e)e^{-\vartheta_2 t}e_2(x, t, \psi, \varphi)\rangle dt + \kappa\|e_3(x, t, \psi, \varphi)\|^{\kappa-2}\langle e_3(x, t, \psi, \varphi), d_3 \triangle e_3(x, t, \psi, \varphi)$

$+ke_2(x, t, \psi, \varphi) - m_e e_3(x, t, \psi, \varphi) - (m_0 - m_e)e^{-\vartheta_3 t}e_3(x, t, \psi, \varphi)\rangle dt + \frac{1}{2}\kappa(\kappa - 1)\xi_1^2(t)\|e_1(x, t, \psi, \varphi)\|^\kappa dt$

$+\frac{1}{2}\kappa(\kappa - 1)\xi_2^2(t)\|e_2(x, t, \psi, \varphi)\|^\kappa dt + \frac{1}{2}\kappa(\kappa - 1)\xi_3^2(t)\|e_3(x, t, \psi, \varphi)\|^\kappa dt + \kappa\|e_1(x, t, \psi, \varphi)\|^{\kappa-2}\langle e_1(x, t, \psi, \varphi),$

$-\xi_1(t) \triangle e_1(x, t, \psi, \varphi)dB_1(t)\rangle + \kappa\|e_2(x, t, \psi, \varphi)\|^{\kappa-2}\langle e_2(x, t, \psi, \varphi), -\xi_2(t) \triangle e_2(x, t, \psi, \varphi)dB_2(t)\rangle +$

$\kappa\|e_1(x, t, \psi, \varphi)\|^{\kappa-2}\langle e_1(x, t, \psi, \varphi), -\xi_3(t) \triangle e_3(x, t, \psi, \varphi)dB_3(t)\rangle\}$

$= \eta e^{\eta t}\|e(x, t, \psi, \varphi)\|^\kappa dt + e^{\eta t}\{-\kappa d_1\|e_1(x, t, \psi, \varphi)\|^{\kappa-2}\|\nabla e_1(x, t, \psi, \varphi)\|^2 - a_e\|e_1(x, t, \psi, \varphi)\|^\kappa$

$-(a_0 - a_e)e^{-\vartheta_1 t}\|e_1(x, t, \psi, \varphi)\|^\kappa - \kappa\beta\|e_1(x, t, \psi, \varphi)\|^{\kappa-2}\langle e_1(x, t, \psi, \varphi), -(u_1(x, t, \psi)e_3(x, t, \psi, \varphi) +$

$u_3(x, t, \varphi)e_1(x, t, \psi, \varphi))\rangle - \kappa d_2\|e_2(x, t, \psi, \varphi)\|^{\kappa-2} \cdot \|\nabla e_2(x, t, \psi, \varphi)\|^2 + \kappa\beta\|e_1(x, t, \psi, \varphi)\|^{\kappa-2}$

$\langle e_1(x, t, \psi, \varphi), -(u_1(x, t, \psi)e_3(x, t, \psi, \varphi) + u_3(x, t, \varphi)e_1(x, t, \psi, \varphi))\rangle - b_e\|e_2(x, t, \psi, \varphi)\|^\kappa$

$-(b_0 - b_e)e^{-\vartheta_2 t}\|e_2(x, t, \psi, \varphi)\|^\kappa - \kappa d_3\|e_3(x, t, \psi, \varphi)\|^{\kappa-2}\|\nabla e_3(x, t, \psi, \varphi)\|^2 + \kappa k\|e_3(x, t, \psi, \varphi)\|^{\kappa-1} \cdot$

$\|e_2(x, t, \psi, \varphi)\| - m_e\|e_3(x, t, \psi, \varphi)\|^\kappa - (m_0 - m_e)e^{-\vartheta_3 t}\|e_3(x, t, \psi, \varphi)\|^\kappa + \frac{1}{2}\kappa(\kappa - 1)\xi_1^2(t)\|e_1(x, t, \psi, \varphi)\|^\kappa$

$+\frac{1}{2}\kappa(\kappa - 1)\xi_2^2(t)\|e_2(x, t, \psi, \varphi)\|^\kappa + \frac{1}{2}\kappa(\kappa - 1)\xi_3^2(t)\|e_3(x, t, \psi, \varphi)\|^\kappa\}dt - \kappa e^{\eta t}\xi_1(t)\|e_1(x, t, \psi, \varphi)\|^\kappa dB_1(t)$

$-\kappa e^{\eta t}\xi_2(t)\|e_2(x, t, \psi, \varphi)\|^\kappa dB_2(t) - \kappa e^{\eta t}\xi_3(t)\|e_3(x, t, \psi, \varphi)\|^\kappa dB_3(t)$

$\leq \eta e^{\eta t}\|e(x, t, \psi, \varphi)\|^\kappa dt + \kappa e^{\eta t}\{\beta M_1\|e_1(x, t, \psi, \varphi)\|^{\kappa-1}\|e_3(x, t\psi, \varphi)\| + \beta M_1\|e_1(x, t, \psi, \varphi)\|^\kappa + k$

$\|e_3(x, t, \psi, \varphi)\|^{\kappa-1}\|e_2(x, t, \psi, \varphi)\| + \frac{1}{2}\kappa(\kappa - 1)\xi_1^2(t)\|e_1(x, t, \psi, \varphi)\|^\kappa + \frac{1}{2}\kappa(\kappa - 1)\xi_2^2(t)\|e_2(x, t, \psi, \varphi)\|^\kappa$

$+\frac{1}{2}\kappa(\kappa - 1)\xi_3^2(t)\|e_3(x, t, \psi, \varphi)\|^\kappa\}dt,$

integrate on both sides of the above inequality and take expectations, at the same time, apply the Young inequality, we get

$$E[e^{\eta t}\|e(x,t,\psi,\varphi)\|^{\kappa}]$$

$$\leq \|e(x,0,\psi,\varphi)\|^{\kappa} + E\int_0^t e^{\eta s}\{\eta\|e(x,s,\psi,\varphi)\|^{\kappa} + (\kappa-1)\|e_1(x,s,\psi,\varphi)\|^{\kappa} + \beta^{\kappa}M_1^{\kappa}\|e_3(x,s,\psi,\varphi)\|^{\kappa}$$

$$+\kappa\beta M_1\|e_1(x,s,\psi,\varphi)\|^{\kappa} + (\kappa-1)\|e_3(x,s,\psi,\varphi)\|^{\kappa} + k^{\kappa}\|e_2(x,s,\psi,\varphi)\|^{\kappa} + \frac{1}{2}\kappa(\kappa-1)\xi_1^2(t)$$

$$\|e_1(x,t,\psi,\varphi)\|^{\kappa} + \frac{1}{2}\kappa(\kappa-1)\xi_2^2(t)\|e_2(x,t,\psi,\varphi)\|^{\kappa} + \frac{1}{2}\kappa(\kappa-1)\xi_3^2(t)\|e_3(x,t,\psi,\varphi)\|^{\kappa}\}ds.$$

Next, we take the supremum on both sides of the above inequality

$$E\sup_{0\leq t\leq T}[e^{\eta t}\|e(x,t,\psi,\varphi)\|^{\kappa}] \leq \|e(x,0,\psi,\varphi)\|^{\kappa} + E\sup_{0\leq t\leq T}M_5\int_0^t e^{\eta s}\|e(x,s,\psi,\varphi)\|^{\kappa}ds, \qquad (17)$$

here
$$M_5 = max\{\eta, (\kappa-1) + \kappa\beta M_1 + \tfrac{1}{2}\kappa(\kappa-1)\xi_1^2, k^{\kappa} + \tfrac{1}{2}\kappa(\kappa-1)\xi_2, (\kappa-1) + \beta^{\kappa}M_1^{\kappa} + \tfrac{1}{2}$$
$$\kappa(\kappa-1)\xi_3^2\}.$$

Based on the Gronwall inequality, we obtain

$$\|e(x,t,\psi,\varphi)\|^{\kappa} \leq \|e(x,0,\psi,\varphi)\|^{\kappa}e^{-\eta t},$$

thereby

$$\lim_{t\to\infty} E\|e(x,t,\psi,\varphi)\|^{\kappa} = 0.$$

Therefore, condition (1) in Lemma 3.6 holds, there exists a stationary distribution for system (8). Next, we prove the uniqueness of stationary distribution, assume that $\bar{\lambda}$ is also a stationary distribution to $\mathcal{W}(x,t)$, there exists some constant $M > 0$, We can get the following result

$$|\lambda(f) - \bar{\lambda}(f)| \leq \int_{\Omega\times\Omega} |P_t f(\psi) - P_t f(\varphi)|\lambda(d\psi)\bar{\lambda}(d\varphi) \leq Me^{-\eta t},$$

when $t \to \infty$, we can get the uniqueness of stationary distribution.

**Remark 6** Theorem 3.7 illustrated the existence and uniqueness of stationary distribution of the solution for the diffusion HBV infection model.

## 4. Numerical simulations

We present the numerical simulation in this section to better understand our results. Based on the Milstein method [25], The system (8) discrete form is as follows:

$$
\begin{aligned}
u_{1(i,j+1)} &= u_{1(i,j)} + [d_1 \frac{u_{1(i+1,j)} - 2u_{1(i,j)} + u_{1(i-1,j)}}{(\triangle x)^2} + \lambda - a_e u_{1(i,j)} - (a_0 - a_e)e^{-\vartheta_1 t}u_{1(i,j)} \\
&\quad - \beta u_{1(i,j)}u_{3(i,j)}] \triangle t - \frac{\xi_1}{\sqrt{2\vartheta_1}}\sqrt{1 - e^{-2\vartheta_1(j\triangle t)}}u_{1(i,j)}\varsigma_j - \frac{\xi_1^2}{4\vartheta_1}(1 - e^{-2\vartheta_1(j\triangle t)})u_{1(i,j)}^2(\varsigma_j^2 - 1)\triangle t,
\end{aligned}
$$

$$
\begin{aligned}
u_{2(i,j+1)} &= u_{2(i,j)} + [d_2 \frac{u_{2(i+1,j)} - 2u_{2(i,j)} + u_{2(i-1,j)}}{(\triangle x)^2} + \beta u_{1(i,j)}u_{3(i,j)} - b_e u_{2(i,j)} - (b_0 - b_e e^{-\vartheta_2 t}) \cdot \\
&\quad u_{2(i,j)}] \triangle t - \frac{\xi_2}{\sqrt{2\vartheta_2}}\sqrt{1 - e^{-2\vartheta_2(j\triangle t)}}u_{2(i,j)}\varsigma_j - \frac{\xi_2^2}{4\vartheta_2}(1 - e^{-2\vartheta_2(j\triangle t)})u_{2(i,j)}^2(\varsigma_j^2 - 1)\triangle t,
\end{aligned}
$$

$$
\begin{aligned}
u_{3(i,j+1)} &= u_{3(i,j)} + [d_3 \frac{u_{3(i+1,j)} - 2u_{3(i,j)} + u_{3(i-1,j)}}{(\triangle x)^2} + ku_{2(i,j)} - m_e u_{3(i,j)} - (m_0 - m_e e^{-\vartheta_3 t})u_{3(i,j)}] \triangle t \\
&\quad - \frac{\xi_3}{\sqrt{2\vartheta_3}}\sqrt{1 - e^{-2\vartheta_3(j\triangle t)}}u_{3(i,j)}\varsigma_j - \frac{\xi_3^2}{4\vartheta_3}(1 - e^{-2\vartheta_3(j\triangle t)})u_{3(i,j)}^2(\varsigma_j^2 - 1)\triangle t,
\end{aligned}
$$

where $\varsigma_j$, $(j = 1, 2, 3)$ are independent Gaussian random variables $N(0, 1)$. We select the $\triangle t = 0.1$, $\triangle x = 0.5$, $a_0 = 0.15$, $b_0 = 2.6$ and $m_0 = 0.35$, other parameter values are chosen in Table 1:

initial value: $(u_{10}(x), u_{20}(x), u_{30}(x)) = (sin\frac{\pi x}{60}, sin\frac{\pi x}{60}, sin\frac{\pi x}{60})$.

### 4.1. The influence of reversion rates for the stationary distribution of the solution

In this section, we consider the stationary distribution of solution of the system (8). In Fig 1, we can see the existence of the stationary distribution of the solution of system (8). The two-dimensional figure on the right shows the changes in time of the solution in different Spaces, and it can be seen that the stationary distribution of the solution is different in different Spaces. The effect of reversion rates on the solution's stationary distribution is depicted in Fig 2. For a more intuitive observation of the effect of the response rate in Fig 2, we present Figs 3–5, as the reversion rates, the amplitude of fluctuation becomes smaller, corresponding to the solution distribution being closer to the normal distribution. On the contrary, the smaller the reversion rates, the stronger the vibration and the more dispersed solutions distribution.

Table 1. Parameter values.

| Parameter | Value | Parameter | Value | Parameter | Value |
|-----------|-------|-----------|-------|-----------|-------|
| $\lambda$ | $10(cells\ ml\ day^{-1})$ [26] | $a_e$ | $0.1(day^{-1})$ [26] | $\xi_1$ | 0.01 [Assumed] |
| $\beta$ | $0.00025(ml\ virion\ day^{-1})$ [26] | $\vartheta_1$ | 0.8[Assumed] | $b_e$ | $2.4(day^{-1})$ [Assumed] |
| k | $50(virion\ cells\ day^{-1})$ [26] | $\xi_2$ | 0.03[Assumed] | $m_e$ | 0.4[Assumed] |
| $\xi_3$ | 0.05[Assumed] | $d_1$ | 0.01[Assumed] | $d_2$ | 0.03[Assumed] |
| $d_3$ | 0.04[Assumed] | $\vartheta_2$ | 0.6[Assumed] | $\vartheta_3$ | 1[Assumed] |

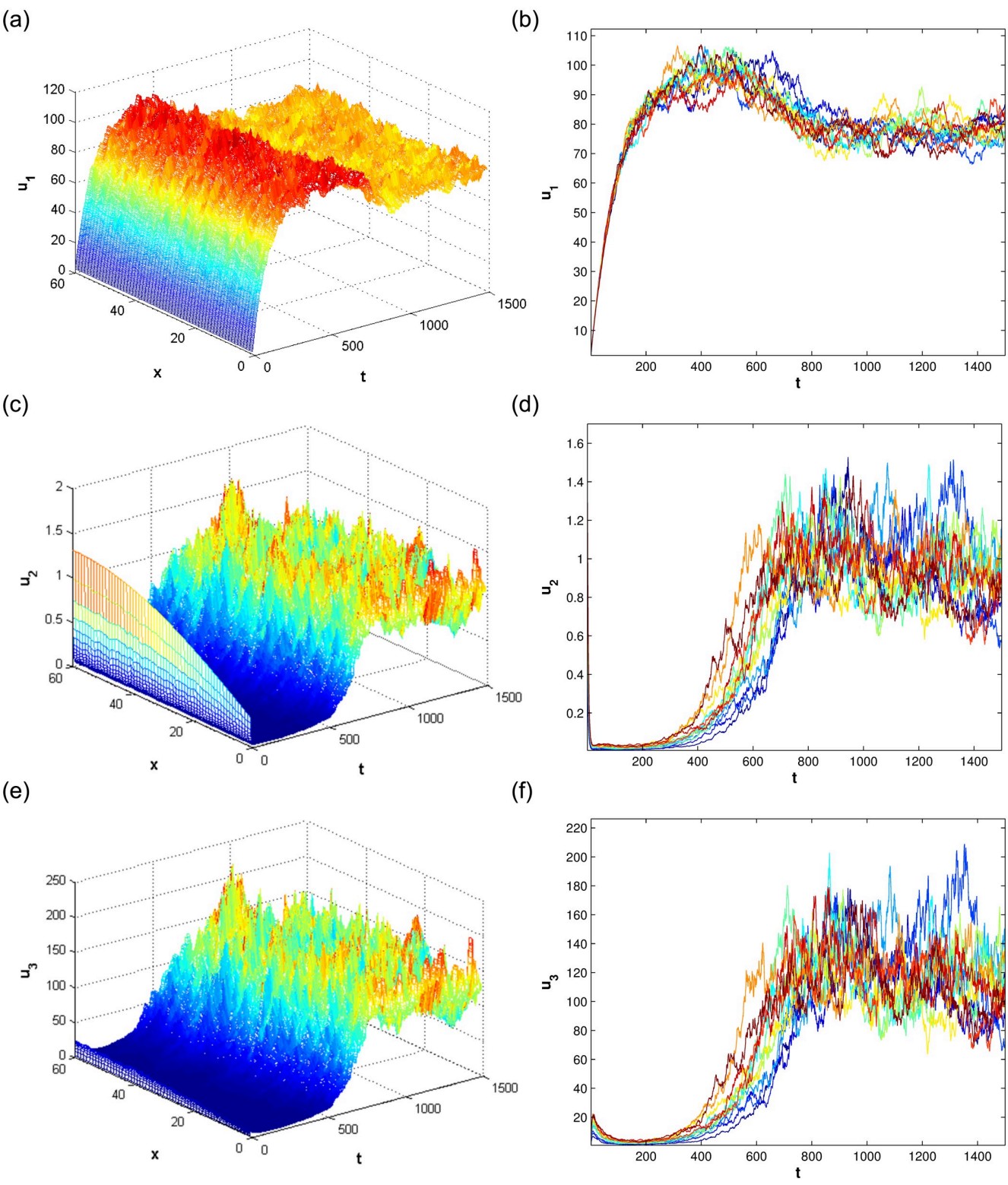

**Fig 1. The stationary distribution of the solution for system (8).**

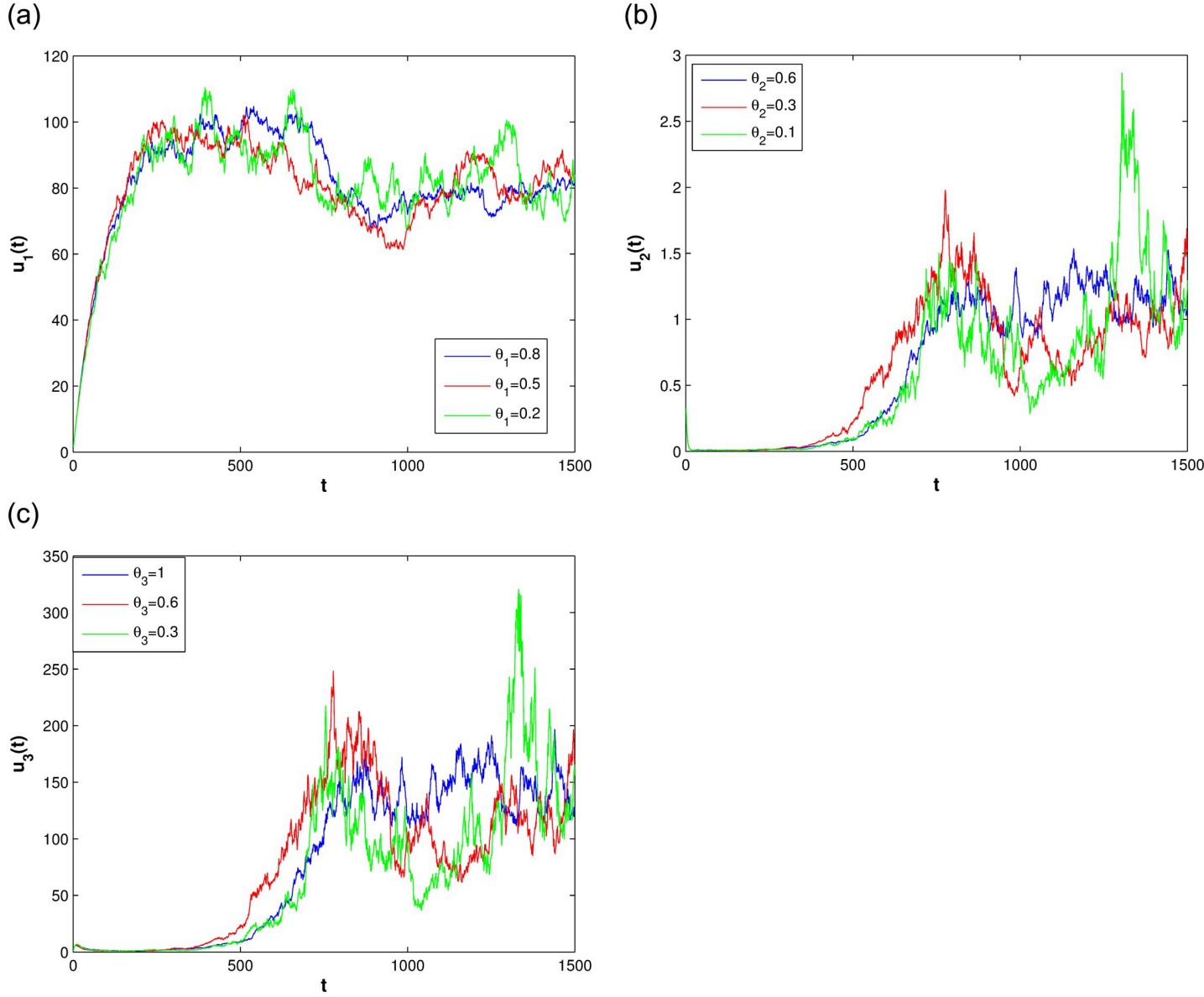

**Fig 2. The impact of difference reversion rates for system (8).**

## 4.2. Impact of noise intensity for stationary distribution of solution

This section considers the influence of noise intensity on the stationary distribution of solutions. The image fluctuation decreases as the noise intensity decreases (Fig 6), for ease of observation, we present the histograms of $u_1$, $u_2$, $u_3$ for each case in Fig 6, and it can be seen that the smaller the noise, the closer the solution is to the normal distribution [see Figs 7–9].

## 5. Conclusions

Mathematical models are regarded as an efficient method when it comes to comprehending how HBV is transmitted. In recent years, many papers have investigated the dynamical behavior of the model, among which we list Din and Li [6], Ge et al. [11], Wu and Zou [16] and

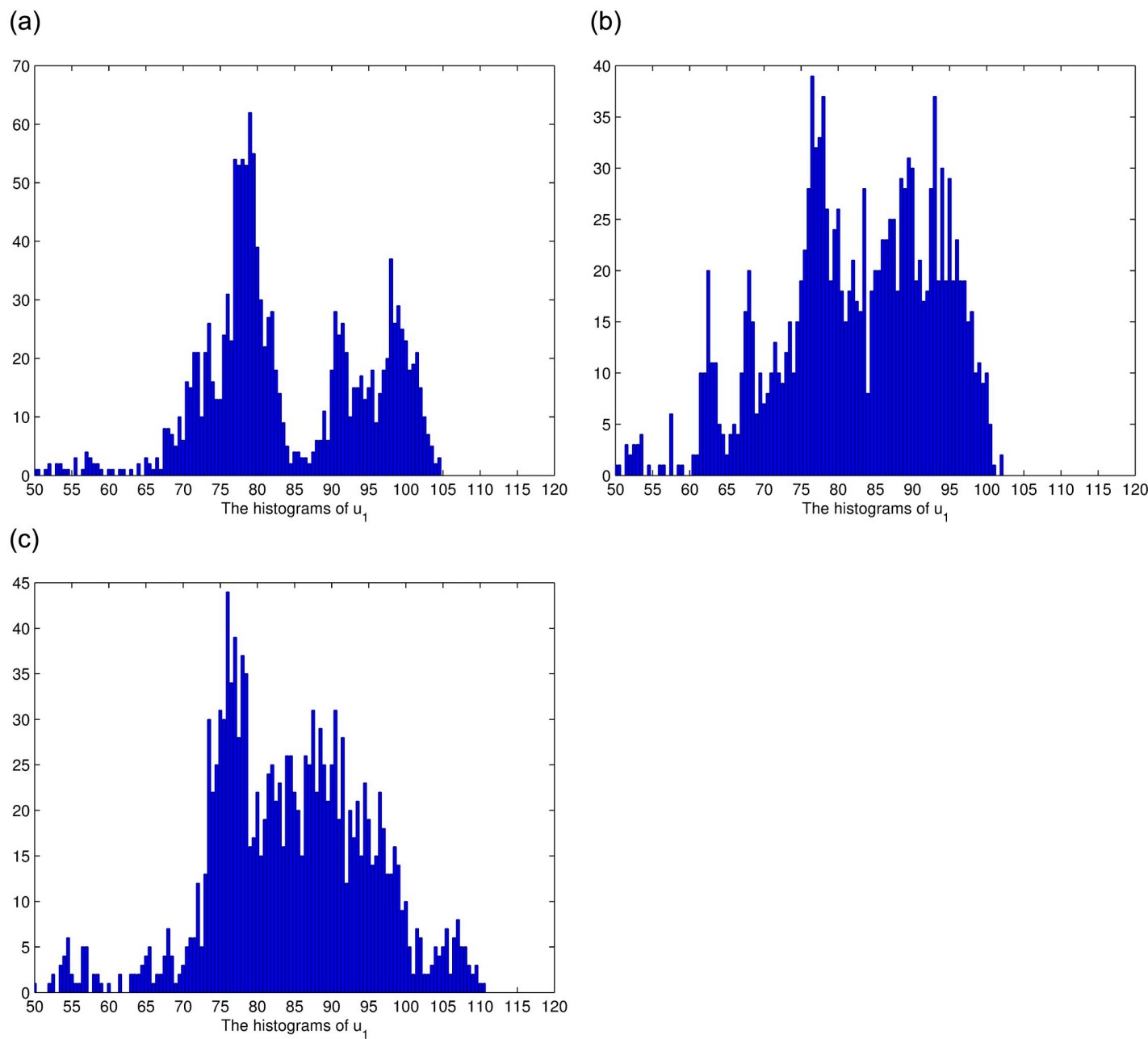

**Fig 3. The histograms of $u_1$ for $\theta_1 = 0.8$, $\theta_1 = 0.5$, $\theta_1 = 0.2$, respectively.**

other related literatures. However, the models in these literature are all derived from ordinary differential equations, or only one that considers the diffusion of cells and viruses, ignoring the simultaneous migration of cells and viruses, that is, spatial diffusion.

This study investigated a stochastic HBV infection model combined with diffusion of cells and viruses and the mean-reverting Ornstein-Uhlenbeck process. We first demonstrate the stationary distribution of the solution to the diffusion model of the HBV infection was shown to exist and be unique under sufficient conditions. The influence of reversion rates and noise

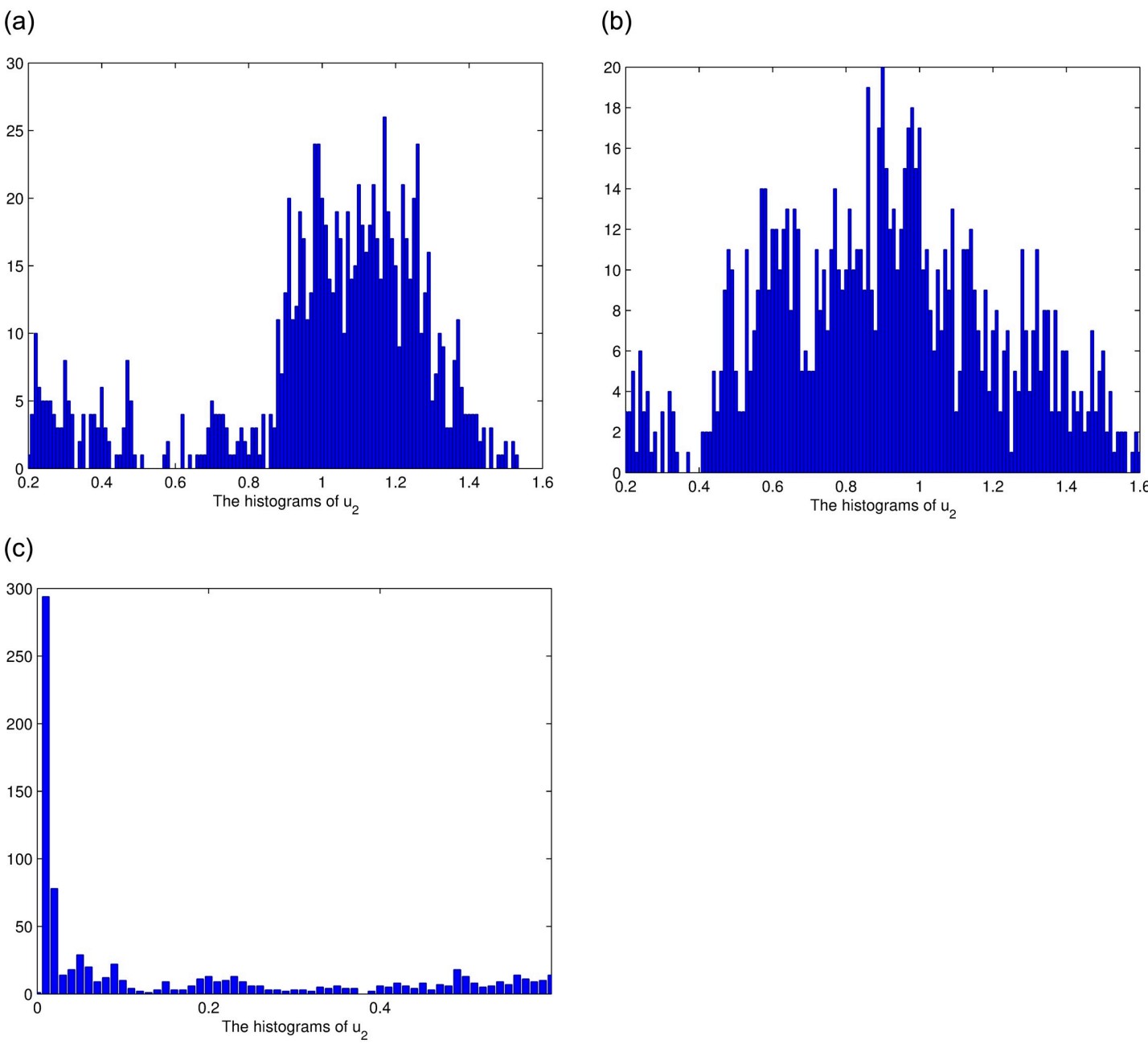

**Fig 4. The histograms of $u_2$ for $\theta_2 = 0.6$, $\theta_2 = 0.3$, $\theta_2 = 0.1$, respectively.**

intensity on the disease is shown, the higher the reversion rates and the smaller the noise, the closer the solution is to the normal distribution. Therefore, increasing the reversion rates and reducing the influence of random factors are beneficial to the treatment of the disease. Meanwhile, the stationary distribution means the disease will persist long-term once infected. Because the system may be disrupted by impulsive perturbations, Markov switching, Lévy jumps, and other random factors, it remains a problem that requires further investigation. We will explore these issues in our future work.

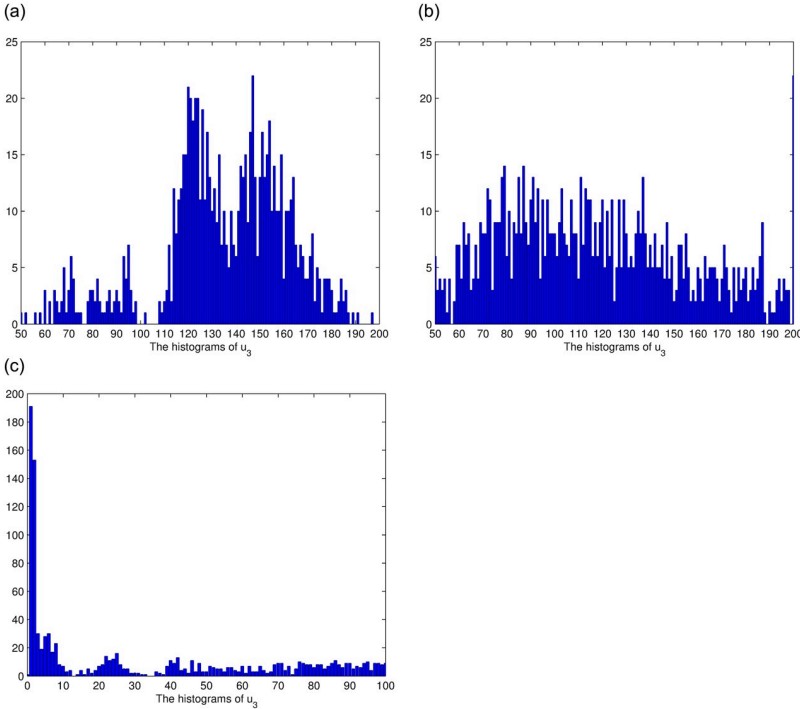

**Fig 5. The histograms of $u_3$ for $\theta_3 = 1$, $\theta_3 = 0.6$, $\theta_3 = 0.3$, respectively.**

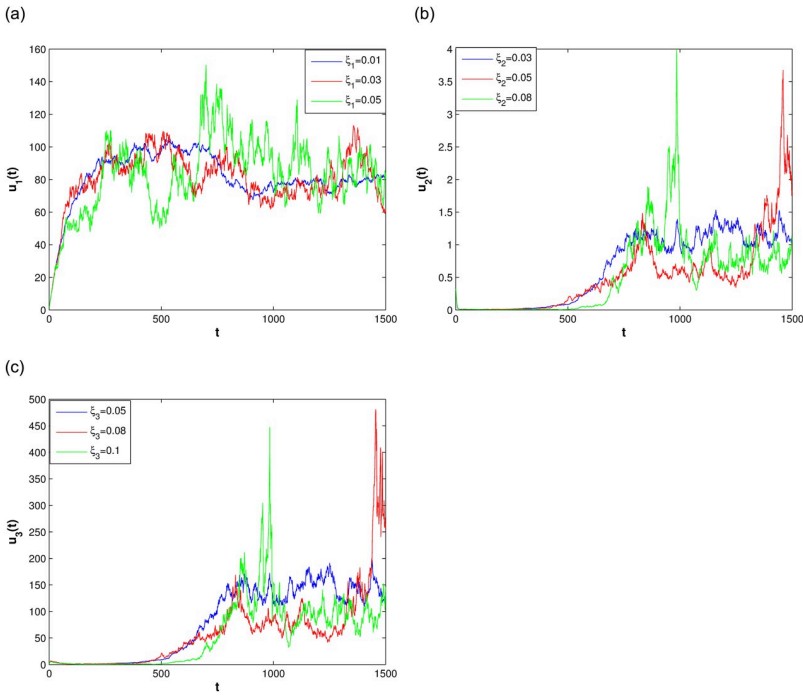

**Fig 6. The impact of difference noise intensity for system (8).**

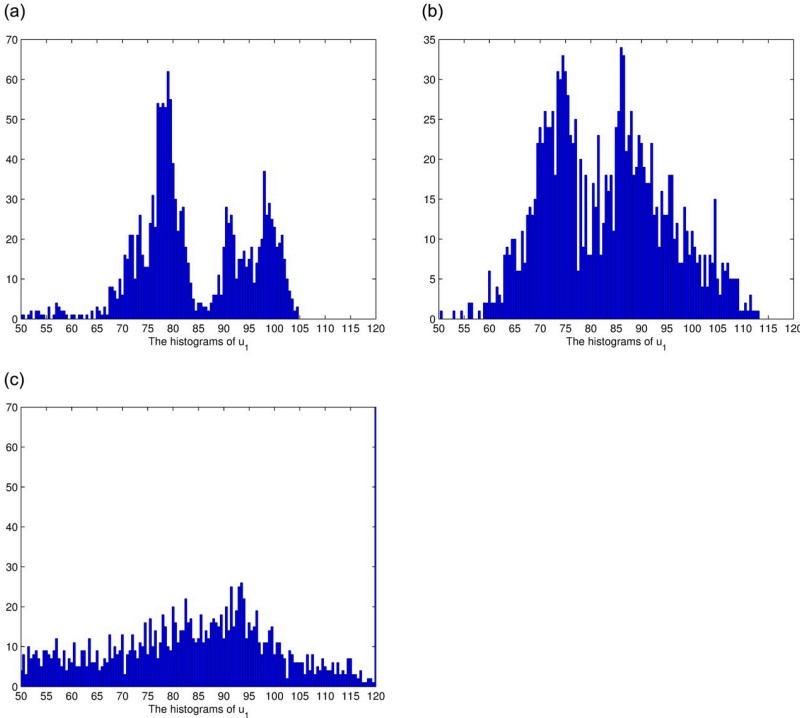

**Fig 7. The histograms of $u_1$ for $\xi_3 = 0.01$, $\xi_3 = 0.03$, $\xi_3 = 0.05$, respectively.**

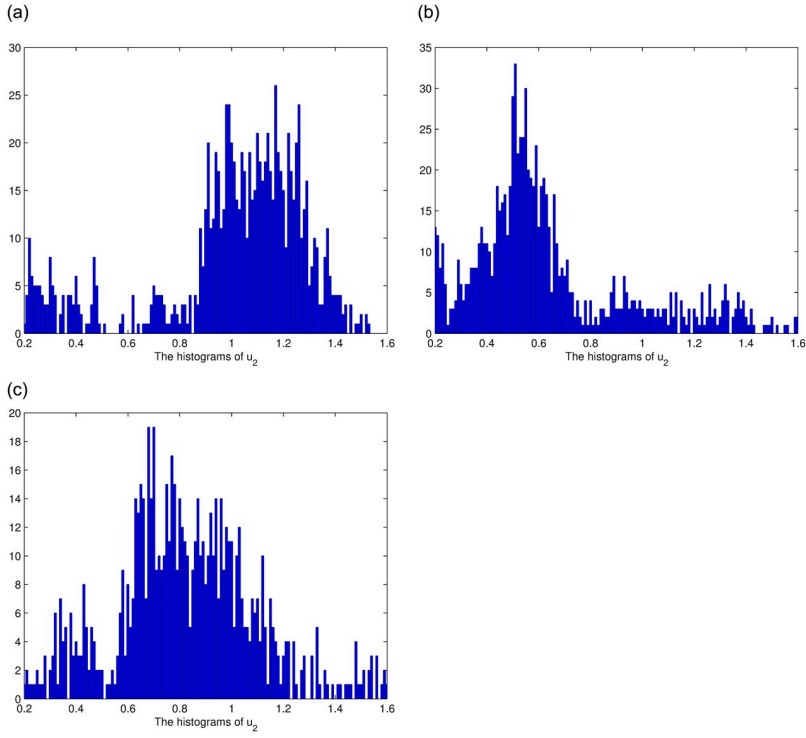

**Fig 8. The histograms of $u_2$ for $\xi_3 = 0.03$, $\xi_3 = 0.05$, $\xi_3 = 0.08$, respectively.**

(a)

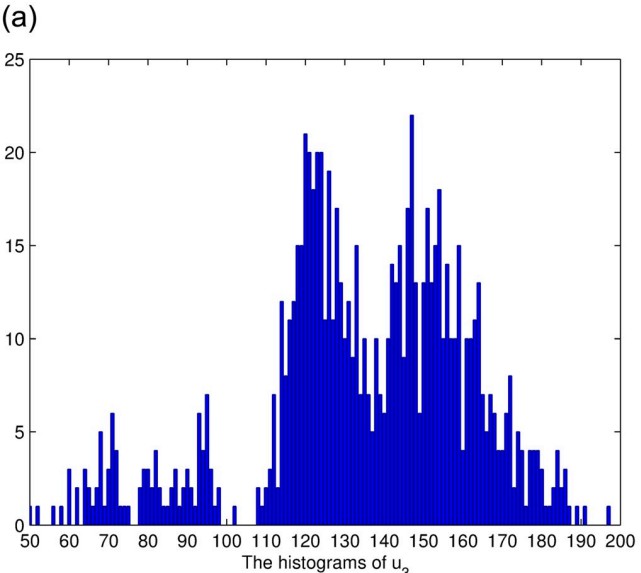

(b)

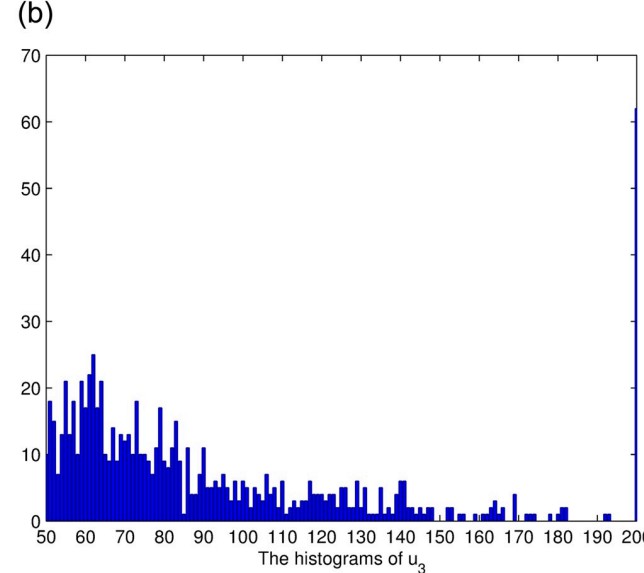

(c)

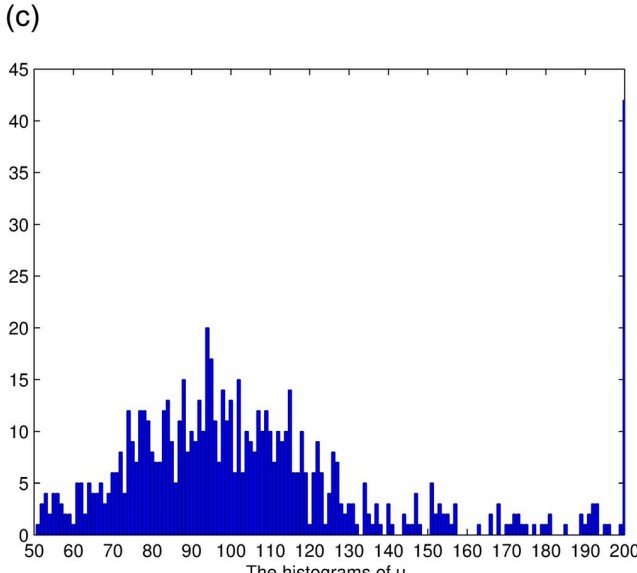

**Fig 9. The histograms of $u_3$ for $\xi_3 = 0.05$, $\xi_3 = 0.08$, $\xi_3 = 0.1$, respectively.**

## Author Contributions

**Writing – original draft:** Guizhen Liang, Kangkang Chang.

**Writing – review & editing:** Zhenyu Zhang.

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
