## [Decision Letter · Decision Letter 0]

2 Jun 2023

PONE-D-23-12836Stationary distribution of a reaction-diffusion hepatitis B virus infection model driven by the Ornstein-Uhlenbeck processPLOS ONE

Dear Dr. Liang,

Thank you for submitting your manuscript to PLOS ONE. After careful consideration, we feel that it has merit but does not fully meet PLOS ONE’s publication criteria as it currently stands. Therefore, we invite you to submit a revised version of the manuscript that addresses the points raised during the review process.

We look forward to receiving your revised manuscript.

Kind regards,

Abdessamad Tridane

Academic Editor

PLOS ONE

Journal Requirements:

"The research was supported in part by the Startup Foundation for Doctors of Xinxiang University

(No.1366020229)."

Reviewers' comments:

Reviewer's Responses to Questions

**Comments to the Author**

1. Is the manuscript technically sound, and do the data support the conclusions?

Reviewer #1: Yes

Reviewer #2: Yes

2. Has the statistical analysis been performed appropriately and rigorously? 

Reviewer #1: Yes

Reviewer #2: N/A

3. Have the authors made all data underlying the findings in their manuscript fully available?

Reviewer #1: Yes

Reviewer #2: Yes

4. Is the manuscript presented in an intelligible fashion and written in standard English?

Reviewer #1: Yes

Reviewer #2: No

5. Review Comments to the Author

Reviewer #1: This paper studies a reaction-diffusion hepatitis B virus (HBV) infection

model based on the mean-reverting Ornstein-Uhlenbeck process. It can be accepted

for the publication after some major revisions.

See the attached file.

Reviewer #2: The authors proposed a reaction-diffusion hepatitis B virus (HBV) infection model based on the mean-reverting Ornstein-Uhlenbeck process. They investigated the qualitative and quantitative behavior of the model.

Comments:

1) In the introduction the authors need to clarify more why the mean-reverting Ornstein-Uhlenbeck process has better characteristics than white noise in biological models.

2) In Page 2; the statement " It was concluded that the solution tends more toward the stationary distribution, the higher the fluctuation and the lower the noise." should be clarified more.

3) What is the biological interpretation for the stationary distribution property?

4) The numerical simulations part should be revised in terms of the following points: a) the discussion for subsections 4.1 \\& 4.2 are not clear and should be improved, b) the captions of all figures should be revised.

5) In Conclusion section, the authors should interpret the main results, place them in context of previous findings.

6) In the introduction, it is recommended to mention the names of authors or simply "In [11,12]...." rather than i.e. Before model (1) "References [11,12]...." ; In addition, the primary goals of the paper can be organized in a better way.

7) Check the statement after (4) "the mean-revertingmeanreverting".

8) The paper needs proofreading.

6. PLOS authors have the option to publish the peer review history of their article (what does this mean?). If published, this will include your full peer review and any attached files.

Reviewer #1: No

Reviewer #2: No

---

## [Author Response · Author response to Decision Letter 0]

18 Jul 2023

Dear editor:

 Thank you for the opportunity to revise my article entitled ”Stationary distribution of a reaction-diffusion hepatitis B virus infection model driven by Ornstein-Uhlenbeck process”(Manuscript ID: PONE-D-23-1283). Through personal negligence, we made some mistakes and wrote some unclear sentences in the original manuscript. The editor and reviewers have pointed out many mistakes and corrected them, for which we express our great appreciation. Please refer to Attachment "Response to reviewers" for specific reply comments

Thank you and best wishes.

Yours sincerely,

Guizhen Liang

---

## [Decision Letter · Decision Letter 1]

12 Sep 2023

Stationary distribution of a reaction-diffusion hepatitis B virus infection model driven by the Ornstein-Uhlenbeck process

PONE-D-23-12836R1

Dear Dr. Liang,

We’re pleased to inform you that your manuscript has been judged scientifically suitable for publication and will be formally accepted for publication once it meets all outstanding technical requirements.

Kind regards,

Abdessamad Tridane

Academic Editor

PLOS ONE

Additional Editor Comments (optional):

Reviewers' comments:

Reviewer's Responses to Questions

**Comments to the Author**

1. If the authors have adequately addressed your comments raised in a previous round of review and you feel that this manuscript is now acceptable for publication, you may indicate that here to bypass the “Comments to the Author” section, enter your conflict of interest statement in the “Confidential to Editor” section, and submit your "Accept" recommendation.

Reviewer #1: All comments have been addressed

Reviewer #2: All comments have been addressed

2. Is the manuscript technically sound, and do the data support the conclusions?

Reviewer #1: Yes

Reviewer #2: Yes

3. Has the statistical analysis been performed appropriately and rigorously? 

Reviewer #1: Yes

Reviewer #2: N/A

4. Have the authors made all data underlying the findings in their manuscript fully available?

Reviewer #1: Yes

Reviewer #2: Yes

5. Is the manuscript presented in an intelligible fashion and written in standard English?

Reviewer #1: Yes

Reviewer #2: Yes

6. Review Comments to the Author

Reviewer #1: (No Response)

Reviewer #2: (No Response)

7. PLOS authors have the option to publish the peer review history of their article (what does this mean?). If published, this will include your full peer review and any attached files.

Reviewer #1: **Yes: **Khalid Hattaf

Reviewer #2: No

---

## [Editor Report · Acceptance letter]

19 Sep 2023

PONE-D-23-12836R1 

Stationary distribution of a reaction-diffusion hepatitis B virus infection model driven by the Ornstein-Uhlenbeck process 

Dear Dr. Liang:

I'm pleased to inform you that your manuscript has been deemed suitable for publication in PLOS ONE. Congratulations! Your manuscript is now with our production department. 

Kind regards, 

on behalf of

Dr. Abdessamad Tridane 

Academic Editor

PLOS ONE